# *Mycobacterium tuberculosis* Evasion of Guanylate Binding Protein-Mediated Host Defense in Mice Requires the ESX1 Secretion System

**DOI:** 10.3390/ijms24032861

**Published:** 2023-02-02

**Authors:** Andrew J. Olive, Clare M. Smith, Christina E. Baer, Jörn Coers, Christopher M. Sassetti

**Affiliations:** 1Department of Microbiology & Molecular Genetics, College of Osteopathic Medicine, Michigan State University, East Lansing, MI 48824, USA; 2Department of Molecular Genetics and Microbiology, Duke University Medical Center, Durham, NC 22710, USA; 3Duke Human Vaccine Institute, Duke University Medical Center, Durham, NC 27710, USA; 4Department of Microbiology and Physiological Systems, University of Massachusetts Medical School, Worcester, MA 01650, USA; 5Department of Immunology, Duke University Medical Center, Durham, NC 22710, USA

**Keywords:** *Mycobacterium tuberculosis*, guanylate binding proteins, interferon-gamma responses, cell-autonomous immunity, immune evasion

## Abstract

Cell-intrinsic immune mechanisms control intracellular pathogens that infect eukaryotes. The intracellular pathogen *Mycobacterium tuberculosis* (*Mtb*) evolved to withstand cell-autonomous immunity to cause persistent infections and disease. A potent inducer of cell-autonomous immunity is the lymphocyte-derived cytokine IFNγ. While the production of IFNγ by T cells is essential to protect against *Mtb*, it is not capable of fully eradicating *Mtb* infection. This suggests that *Mtb* evades a subset of IFNγ-mediated antimicrobial responses, yet what mechanisms *Mtb* resists remains unclear. The IFNγ-inducible Guanylate binding proteins (GBPs) are key host defense proteins able to control infections with intracellular pathogens. GBPs were previously shown to directly restrict *Mycobacterium bovis* BCG yet their role during *Mtb* infection has remained unknown. Here, we examine the importance of a cluster of five GBPs on mouse chromosome 3 in controlling Mycobacterial infection. While *M. bovis* BCG is directly restricted by GBPs, we find that the GBPs on chromosome 3 do not contribute to the control of *Mtb* replication or the associated host response to infection. The differential effects of GBPs during *Mtb* versus *M. bovis* BCG infection is at least partially explained by the absence of the ESX1 secretion system from *M. bovis* BCG, since *Mtb* mutants lacking the ESX1 secretion system become similarly susceptible to GBP-mediated immune defense. Therefore, this specific genetic interaction between the murine host and *Mycobacteria* reveals a novel function for the ESX1 virulence system in the evasion of GBP-mediated immunity.

## 1. Introduction

Eukaryotic cells control intracellular pathogens using a variety of cell intrinsic immune pathways [1]. These innate mechanisms allow the cell to rapidly detect, target and destroy invading pathogens, preventing the spread of an infection. The immune pathways controlling innate immunity arose early in the evolution of the eukaryota, providing ample time for the selection of pathogens that express mechanisms to bypass cell-autonomous immunity [2]. This long-term arms race has produced a myriad of interactions between immune effectors and pathogen countermeasures that determine the outcome of an infection.

*Mycobacterium tuberculosis* (*Mtb*) has become highly adapted to its human host, as it spread globally along with human migrations over tens of thousands of years [3,4]. As much as one-third of the human population has been exposed to *Mtb*, which causes a persistent infection than can last for years, or even decades, despite a robust immune response that eradicates less pathogenic mycobacteria [5]. While immunity controls *Mtb* growth and prevents disease in most individuals, a subset will develop active tuberculosis (TB), a disease that kills an estimated 1.8 million each year [6]. Disease progression is influenced by a variety of genetic and environment factors, but ultimately is determined by the interplay between host immunity and bacterial virulence systems [5,7]. 

In the host, the development of a robust T cell response and the production of the cytokine IFNγ are important for the control of TB infection and disease [5,8]. Humans with inherited mutations affecting either the development or expression of this response are highly susceptible to mycobacterial infections including TB [9]. This susceptibility is faithfully modeled in mice, where the loss of IFNγ signaling promotes disease by at least two distinct mechanisms [10,11]. IFNγ is an important immunomodulatory cytokine, and its loss results in uncontrolled IL1 production and neutrophil recruitment, driving both bacterial replication and tissue damage [10,12]. Perhaps more importantly, IFNγ stimulates cell-intrinsic immune pathways in phagocytes, which is critical for the control of intracellular bacterial growth [13,14]. Thus, IFNγ is a pleotropic cytokine that controls direct antimicrobial resistance and disease tolerance, both of which are essential to survive *Mtb* infection. 

While many of the immunomodulatory effects of IFNγ have been elucidated, the IFNγ-induced factors that control *Mtb* replication remain comparatively obscure. IFNγ-induced oxygen and nitrogen radical generation limits *Mtb* replication in macrophages ex vivo but appear to serve a limited antimicrobial role in the intact animal [14,15,16]. Instead, a subset of IFNγ-inducible cell-intrinsic immune mechanisms, known as Guanylate binding proteins (GBPs), target and disrupt the intracellular niche required for a number of pathogens to grow [2,17,18,19,20,21]. Macrophages lacking GBPs 1, 6, 7 or 10 fail to control the growth of *M. bovis* BCG, an attenuated vaccine strain that is closely related to *Mtb* [22]. Similarly, mice lacking GBP1 or the cluster of five GBP proteins on chromosome 3 are more susceptible to intravenous BCG challenge [22,23]. While the mechanism of BCG control remains unclear, GBPs are known to bind to pathogen containing vacuoles in other infections and recruit additional effector molecules [17,24]. The outcome of this recognition can be the direct restriction of pathogen growth and alterations in cytokine production. While these observations suggest that GBPs may be an important mediator of IFNγ-mediated control of pathogenic mycobacteria, their role in during *Mtb* infection has remained untested. 

BCG was attenuated for use as a vaccine via long term serial passage, and as a result, it interacts with the macrophage quite differently from *Mtb* [25]. Most notably, the primary genetic lesion responsible for the attenuation of BCG is a deletion that disrupts the ESX1 type VII secretion system [26]. ESX1 is specialized protein secretion complex that contributes to pathogen replication by remodeling its intracellular environment [27]. ESX1 is responsible for the disruption of the phagosomal membrane, the activation of multiple cytosolic immune sensing pathways, and the stimulation of both cytokine secretion and autophagy [28]. How ESX1 alters other aspects of cell-intrinsic bacterial control remains to be determined. 

To understand the mechanisms of IFNγ-mediated protection we investigated the role of GBPs during both BCG and *Mtb* infection. Specifically, we examined the function of a cluster of five GBP proteins that are encoded in a single locus on mouse chromosome 3 [29]. While these GBPs restricted *M. bovis* BCG replication, we found no contribution of these proteins in the control of virulent *Mtb* infection. The discrepant effects of GBPs during BCG and *Mtb* infection could be attributed to differential ESX1 function in these two pathogens. ESX1-deficient strains of *Mtb* were better controlled by GBP-mediated immunity, while GBP deficiency had no effect on either the growth of ESX1 expressing *Mtb* or the immune response to this virulent strain. Together, these observations indicate that the ESX1 system plays an essential role in the evasion of GBP-mediated cell intrinsic immunity.

## 2. Results

### 2.1. Mycobacterium Bovis BCG Growth Is Restricted by the Chromosome 3 Guanylate Binding Protein Cluster

The 10 murine GBP genes are encoded in two clusters on chromosomes 3 and 5 [1]. To begin to address the potential redundancy between these genes, we used mice lacking the entire chromosome 3 cluster that contains the Gbp1 and seven genes that were previously implicated in BCG control, along with GBPs 2, 3, and 5 [29]. To determine if these GBPs contribute to control of mycobacterial infection, bone marrow-derived macrophages (BMDMs) from wild type and *Gbp^chr3−/−^* mice were infected with *Mycobacterium bovis* BCG Figure 1A. IFNγ-treatment, which induces GBP expression, reduced the intracellular bacterial burden in wild type BMDMs. In contrast, the *Gbp^chr3−/−^* deletion significantly reduced the ability of macrophages to control BCG upon IFNγ activation. Thus, GBPs on chromosome 3 contribute to IFNγ-mediated restriction of BCG in macrophages.

We next examined if GBPs controlled BCG infection in the context of an intact immune response. Wild type, *IFNγR^−/−^* and *Gbp^chr3−/−^* mice were infected intravenously with BCG. When CFU were quantified in the spleen 50 days later we observed 20-fold more BCG in *IFNγR^−/−^* mice and five-fold more BCG in *Gbp^chr3−/−^* mice compared to wild type controls Figure 1B. The increased susceptibility of *IFNγR^−/−^* compared to *Gbp^chr3−/−^* mice indicated that the function of these GBPs accounted for some but not all of the protective effect of IFNγ. Together these data show that the GBPs on chromosome 3 are able to restrict the growth of BCG in IFNγ-stimulated macrophages and in the intact animal, which is consistent with the previously described roles of GBPs in immunity to BCG [22,23].

### 2.2. Chromosome 3 GBP Cluster Does Not Control Mycobacterium tuberculosis Growth in Macrophages

To determine if the chromosome 3 GBPs restricted the intracellular growth of virulent *Mtb*, in addition to BCG, we quantified the growth of *Mtb* strain H37Rv in BMDMs from wild type, *IFNγR^−/−^* and *Gbp^chr3−/−^* animals in the presence and absence of IFNγ Figure 2A. We observed no differences in the uptake between genotypes four hours following infection. IFNγ priming reduced the intracellular *Mtb* growth by 3–4 fold in wild type macrophages, but not *IFNγR^−/−^* BMDMs at 5 days post infection (dpi). In contrast to BCG, we observed no difference in *Mtb* growth in *Gbp^chr3−/−^* BMDMs compared to wild type macrophages in the presence or absence of IFNγ.

To ensure that the relatively insensitive CFU-based intracellular growth assay did not mask subtle effects of GBP expression on bacterial fitness, we used a fluorescent live/dead reporter as an orthologous method to measure *Mtb* intracellular viability by flow cytometry. This reporter expresses a constitutive mEmerald and an anhydrotetracycline (aTc)-inducible tagRFP. BMDMs from wild type, *IFNγR^−/−^* and *Gbp^chr3−/−^* mice were infected with the Live/Dead *Mtb* reporter and left untreated or were stimulated with IFNγ. Four days later aTc was added to induce tagRFP expression in viable intracellular bacteria. The following day, cells were analyzed by flow cytometry and the mean fluorescence intensity (MFI) of tagRFP in infected mEmerald+ cells was quantified Figure 2B,C. Similar to the CFU analysis, IFNγ activation reduced the intensity of tagRFP to a similar extent in both wild type and *Gbp^chr3−/−^* BMDMs, and this reduction depended on the IFNγ receptor. We further quantified cell viability in all genotypes and observed no significant differences between wild type and *Gbp^chr3−/−^* BMDMs five days following infection Figure 2D. Thus, we were not able to detect a role for the chromosome 3 GBP cluster in the IFNγ mediated restriction of virulent *Mtb* in macrophages. 

### 2.3. Chromosome 3 GBPs Have No Effect on Mtb Infection in Intact Mice

To assess the role of the chromosome 3 GBPs in the more complex setting of the intact animal, we infected wild type, *IFNγR^−/−^* and *Gbp^chr3−/−^* mice with *Mtb* by low dose aerosol and quantified bacteria in the lungs at four and five weeks following infection Figure 3A,B. *IFNγR^−/−^* mice harbored more *Mtb* than wild type controls at both time points in the lung and the spleen. In contrast, the *Mtb* burdens in *Gbp^chr3−/−^* mice were indistinguishable from wild type animals at all timepoints. At 110 days after infection Figure 3A,B, we continued to find no difference in CFU in the lungs between wild type and *GBP^chr3−/−^* mice. *IFNγR^−/−^* mice required euthanasia within six weeks of infection and were not included in this late time point.

To examine more subtle differences in the extent of infection, wild type, IFNγR^−/−^ and Gbp^chr3−/−^ mice were infected with a well-characterized fluorescent H37Rv strain by low-dose aerosol [30]. Twenty-eight days later, the number of YFP+ cells in the lung environment were determined by flow cytometry Figure 3C. Similar to our CFU results, we found IFNγR mice contained over 10 times more infected cells than wild type mice while *Gbp^chr3−/−^* mice showed no significant difference in the number of infected cells per lung Figure 3D.

We next examined if there were differences in *Gbp^chr3−/−^* mice infected intravenously since this route matched the BCG infections where the role of GBPs was evident Figure 1B and [22]. Following intravenous infection with 10^6^
*Mtb*, bacterial levels in the spleen were quantified 10 and 28 days later. Similar to the aerosol infection results, we saw no difference between *Gbp^chr3−/−^* and wild type animals, while *IFNγR^−/−^* animals had 10 times more *Mtb* growth in the spleen Figure 3E. Thus, unlike the loss of IFNγR, the loss of chromosome 3 GBPs does not affect *Mtb* growth in the lungs or spleen, regardless of infection route, suggesting no major function of these five GBPs in controlling antimicrobial resistance to *Mtb* in mice. 

### 2.4. Gbp^chr3−/−^ Mice Continue to Regulate Inflammatory Responses to Mtb Infection

IFNγ protects against disease both by restricting bacterial growth and by inhibiting tissue-damaging inflammation [11]. To determine if the chromosome 3 GBPs contribute to the latter immunoregulatory function of IFNγ, we profiled the immune responses of infected macrophages and mice. Wild type and *GBP^chrm3−/−^* BMDMs were treated with IFNγ then infected with *Mtb* and the following day supernatants were harvested. IFNγ signaling showed the expected inhibitory effect on IL-1β secretion, but the chromosome3 Gbp deletion had no effect on either cytokine or cell viability Figure 4A [12]. We next profiled the host response in the lungs of mice to determine if chromosome 3 GBPs modulated the recruitment of immune cells or the production of cytokines during *Mtb* infection. Wild type, *IFNγR^−/−^* and *Gbp^chr3−/−^* mice were infected with *Mtb* and four weeks later we quantified the immune populations in the lungs by flow cytometry Figure 4B. While we observed the previously described increase in neutrophil recruitment in *IFNγR^−/−^* mice, we observed no differences in any myeloid or lymphoid derived cells that were examined between Gbp^chr3−/−^ and wild type mice [10]. We also quantified IL1β and TNFα in the lungs from these infected animals Figure 4C. The production of these cytokines in wild type and *GBP^chrm3−/−^* mice were indistinguishable while *IFNγR^−/−^* mice showed an increase in IL1β. Thus, we were unable to detect a role for GBPs on chromosome 3 in controlling the host response during virulent *Mtb* infection.

### 2.5. ESX1 Is Required for Mtb to Evade GBPs of Chromosome 3

The reduced virulence of BCG compared to *Mtb* has been largely attributed to the loss of ESX1 function in BCG. As a result, we hypothesized ESX1 function in virulent *Mtb* provides resistance to GBP-mediated immunity, and its loss renders BCG susceptible to this mechanism. To test this hypothesis, we determined whether abrogation of ESX1 function in *Mtb* would result in a strain that was susceptible to GBP-mediated control. Wild type and *Gbp^chr3−/−^* macrophages were infected with H37Rv or two isogenic mutants each lacking a gene that is necessary for ESX1 function, Δ*espA* or Δ*eccb1*. We observed that ESX1 mutants displayed reduced intracellular growth compared to H37Rv in both IFNγ stimulated and unstimulated wild type macrophages, confirming the attenuation of these mutants Figure 5A. When we compared wild type and *Gbp^chr3−/−^* BMDMs, we found that the loss of GBP function had no effect on H37Rv, while it significantly reduced the ability of IFNγ to control the growth of both ESX1 mutants.

To confirm the specific effect of chromosome 3 GBPs on ESX1 mutants in the setting of intact immunity, we conducted a competitive infection of wild type and *Gbp^chr3−/−^* mice with an equivalent number of H37Rv and Δ*eccb1* bacteria. Four weeks later we quantified the competitive index of each bacterial strain in the lungs Figure 5B As anticipated, the ratio of two differentially marked H37Rv strains stayed constant throughout the infection. While selection in wild type mice resulted in almost 100-fold underrepresentation of the Δ*eccb1* mutant compared H37Rv, the fitness defect of the ∆*eccB1* mutant was significantly reduced in the GBP^chr3−/−^ mice Figure 5B. These findings demonstrate that ESX1 deficiency is sufficient to render *Mtb* susceptible to GBP-mediated immunity.

## 3. Discussion

Understanding the immune mechanisms that restrict the intracellular growth of *Mtb* is essential for the rational design of interventions. Initial observations demonstrating a role for GBPs in the control of *BCG* growth suggested that this pathway might represent an important component of IFNγ -mediated immunity to *Mtb* [22,23]. However, while we were able detect the previously-described role for the chromosome 3 GBPs in immunity to BCG, we found that these proteins have no effect during *Mtb* infection. By attributing this difference to the ESX1 locus that is present in *Mtb* but not BCG, we discovered a specific role for ESX1 in overcoming GBP-mediated defenses. 

Our findings question the importance of GBPs in the control of ESX1-expressing *Mtb*. While our results are strictly in the mouse model, evidence in humans also suggests GBPs may not effectively control TB progression. For example, the high expression of a subset of GBPs is predictive of patients that are more likely to progress to active disease [31]. However, it is important to note that it remains possible that the chromosome 5 GBPs, or human-specific GBP functions, can overcome ESX1-mediated bacterial defenses. In addition, our findings do not rule out an important role for GBPs in resistance to non-tuberculous mycobacteria (NTM). While all mycobacteria express ESX paralogs, many pathogenic NTM do not possess a clear ortholog of ESX1, suggesting that they may remain susceptible to GBP immunity [32]. 

How the ESX1 type VII secretion system allows *Mtb* to evade restriction by GBPs remains to be investigated. To date, the type III secretion system effector protein IpaH9.8 from *Shigella flexneri* is the only other described GBP antagonist [33,34,35,36]. In the cytosol of mammalian cells, where *S. flexneri* replicates, IpaH9.8 targets GBP1 for degradation thereby interfering with direct GBP binding to the *Shigella* outer membrane, a process by which GBP1 disrupts the function of a membrane-bound *Shigella* virulence factors required for actin-based motility and bacterial dissemination [33,34,35,37]. We speculate that ESX1 likely functions by either modulating GBPs directly or by inhibiting downstream antimicrobial functions that are dependent on GBPs. If ESX1 modulates GBP expression or post-translational modifications that alter GBP stability it will function similarly to IpaH9.8. If GBP expression and stability is unchanged in the presence of ESX1, then GBP-mediated restriction may be inhibited by evading the antimicrobial mechanisms that occur after GBP translocation to *Mycobacteria*-containing vacuoles. Future work will need to carefully dissect GBP expression and localization dynamics in Mtb infected cells to define the mechanisms for ESX1-mediated antagonism of GBP function.

While IFNγ is unquestionably important to survive *Mtb* infection, the IFNγ-mediated pathways that directly control the intracellular replication of *Mtb* remain surprisingly unclear. Our results add to a growing list of direct antimicrobial pathways that are ineffective during *Mtb* infection. The IFNγ-mediated production of nitric oxide, reactive oxygen species and itaconate kills many pathogens; however, these mechanisms appear to play a small role in directly controlling *Mtb* growth in vivo [30,38,39]. Instead, these mediators are required to inhibit persistent inflammation and to prevent disease progression. In addition, the IFNγ-regulated immunity related GTPases (IRG) family protein, Irgm1, was originally described to target the *Mtb* containing vacuole to control pathogen growth, but recent evidence has questioned whether Irgm1 targets *Mtb* phagosomes [40,41]. Instead, disruption in the balance between Irgm proteins is predominantly responsible for the phenotype seen in Irgm1-deficient mice during *Mtb* infection [42]. While other pathways have been suggested to play a role in IFNγ-mediated control, including the production of Cathepsins, the role of these mediators in protection in vivo remains unclear [43]. Overall, our findings add GBP-mediated immunity to the list of IFNγ dependent host defense programs to which *Mtb* has evolved specific counter immune mechanisms blunting the effectiveness of these antimicrobial effectors and thus driving pathogen persistence and disease. 

## 4. Materials and Methods

### 4.1. Ethics Statement

Mouse studies were performed in strict accordance using the recommendations from the Guide for the Care and Use of Laboratory Animals of the National Institutes of Health and the Office of Laboratory Animal Welfare. Mouse studies were performed using protocols approved by the Institutional Animal Care and Use Committee (Number A3306-01).

### 4.2. Mice

C57BL/6J (Stock # 000664) and IFNγR^−/−^ (Stock # 003288) mice were purchased from the Jackson Laboratory. The Gbp^chr3−/−^ mice were previously described [29]. All knockout mice were housed and bred under specific pathogen-free conditions and in accordance with the University of Massachusetts Medical School IACUC guidelines. All animals used for experiments were 6–12 weeks. 

### 4.3. Bacterial Strains

Wild type *M. tuberculosis* strain H37Rv was used for all studies unless indicated. This strain was confirmed to be PDIM-positive. The espA (Rv3616c) deletion strain (Δ*espA*) was a gift from Dr. Sarah Fortune [44]. The eccb1 (Rv3869) deletion strain (Δ*eccb1*) was constructed in the H37Rv parental background using the ORBIT method as described previously [45]. The live/dead strain was built by transforming the live/dead vector (pmV261 hsp60::mEmerald tetOtetR::TagRFP) into H37Rv and selected with hygromycin. Protein expression was confirmed via fluorescence microscopy and flow cytometry. H37Rv expressing msfYFP has been previously described and the episomal plasmid was maintained with selection in Hygromycin B (50 ug/mL) added to the media [30]. *Mycobacterium bovis* BCG Danish Strain 1331 (Statens Serum Institute, Copenhagen, Denmark) was used for all BCG infection studies. Prior to infection bacteria were cultured in 7H9 medium containing 10% oleic albumin dextrose catalase growth supplement (OADC) enrichment (Becton Dickinson Franklin Lakes, NJ USA) and 0.05% Tween 80. 

### 4.4. Mouse Infection

For low dose aerosol infections (50–150 CFU), bacteria were resuspended in phosphate-buffered saline containing tween 80 (PBS-T). Prior to infection bacteria were sonicated then delivered via the respiratory route using an aerosol generation device (Glas-Col Terre Haute IN, USA). For mixed infections, bacteria were prepared then mixed 1:1 before aerosol infection. To determine CFU, mice were anesthetized via inhalation with isoflurane (Piramal Mumbai, India) and euthanized via cervical dislocation, the organs aseptically removed and individually homogenized, and viable bacteria enumerated by plating 10-fold serial dilutions of organ homogenates onto 7H10 agar plates. Plates were incubated at 37 °C, and bacterial colonies counted after 21 days. Both male and female mice were used throughout the study and no significant differences in phenotypes were observed between sexes. 

### 4.5. Flow Cytometry

Lung tissue was harvested in DMEM containing FBS and placed in C-tubes (Miltenyi Bergisch Gladbach Germany). Collagenase type IV/DNaseI was added and tissues were dissociated for 10 s on a GentleMACS system (Miltenyi Bergisch Gladbach Germany). Tissues were incubated for 30 min at 37 °C with oscillations and then dissociated for an additional 30 s on a GentleMACS. Lung homogenates were passaged through a 70-micron filter or saved for subsequent analysis. Cell suspensions were washed in DMEM, passed through a 40-micron filter and aliquoted into 96-well plates for flow cytometry staining. Non-specific antibody binding was first blocked using Fc-Block. Cells were then stained with anti-Ly-6G Pacific Blue, anti-CD11b PE, anti-CD11c APC, anti-Ly-6C APC-Cy7, anti-CD45.2 PercP Cy5.5, anti-CD4 FITC, anti-CD8 APC-Cy7, anti-B220 PE-Cy7 (Biolegend San Diego, CA USA). Live cells were identified using fixable live dead aqua (Life Technologies Carlsbad CA, USA). For infections with fluorescent H37Rv, lung tissue was prepared as above but no antibodies were used in the FITC channel. All of these experiments contained a non-fluorescent H37Rv infection control to identify infected cells. Cells were stained for 30 min at room temperature and fixed in 1% Paraformaldehyde for 60 min. All flow cytometry was run on a MACSQuant Analyzer 10 (Miltenyi, Bergisch Gladbach Germany) and was analyzed using FlowJo Version 9 (Tree Star).

### 4.6. Bone Marrow-Derived Macrophage Generation 

To generate bone marrow derived macrophages (BMDMs), marrow was isolated from femurs and tibia of age and sex matched mice as previously described [38]. Cells were then incubated in DMEM (Sigma St. Louis MO, USA) containing 10% fetal bovine serum (FBS) and 20% L929 supernatant. Three days later media was exchanged with fresh media and seven days post-isolation cells were lifted with PBS-EDTA and seeded in DMEM containing 10% FBS for experiments. 

### 4.7. Macrophage Infection

*Mtb* or *Mycobacterium bovis*-BCG were cultured in 7H9 medium containing 10% oleic albumin dextrose catalase growth supplement (OADC) enrichment (Becton Dickinson, Franklin Lakes, NJ USA) and 0.05% Tween 80. Before infection cultures were washed in PBS-T, resuspended in DMEM containing 10%FBS and centrifuged at low speeds to pellet clumps. The supernatant was transferred to a new tube to ensure single cells. Multiplicity of infection (MOI) was determined by optical density (OD) with an OD of 1 being equivalent to 3 × 10^8^ bacteria per milliliter. Bacteria were added to macrophages for 4 h then cells were washed with PBS and fresh media was added. At the indicated time points supernatants were harvested for cytokine analysis and the cells were processed for further analysis. For cytokine treatments cells were treated with the indicated concentrations of IFNγ (Peprotech, Cranbury, NJ, USA) or vehicle control 4 h following infection and maintained in the media throughout the experiment. For CFU experiments at the indicated timepoints 1% saponin was added to each well and lysates were serially diluted in PBS. 05% Tween and plated on 7H10 agar and colonies were counted 21–28 days later. For the Live/Dead reporter experiments, Anhydrotetracycline (aTc) (Cayman Chemical Ann Arbor, MI USA) was added to a final concentration of 500 ug/mL 24 h before cells were lifted, fixed in 1% Paraformaldehyde and analyzed on a MacsQuant VYB analyzer. 

### 4.8. Cytokine Quantification by ELISA and Cell Death Quantification

Murine cytokine concentrations in culture supernatants and cell-free lung homogenates were quantified using commercial enzyme-linked immunosorbent assay (ELISA) kits (R&D Minneapolis, MN USA). All samples were normalized for total protein content. Cell viability was quantified using a Cell Titer Glo kit (Promega Madison, WI USA) in short, cells were lysed with Cell titer glo lysis and mixed with substrate. Luminescence was then measured on a Tecan Spark 20M plate reader.

### 4.9. Statistics

GraphPad Prism version 7 was used for all statistical analysis. Unless otherwise indicated one-way ANOVA with a tukey correction was used to compare each condition to each genotype. 

## Figures and Tables

**Figure 1 ijms-24-02861-f001:**
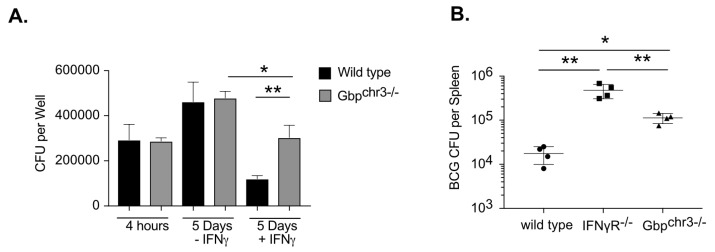
Guanylate binding proteins contribute to control of *Mycobacterium Bovis* BCG infection. (**A**) BMDMs from wild type or *Gbp^chr3−/−^* mice were infected with *M. Bovis* BCG for 4 h then washed with fresh media in the presence or absence of IFNγ. Five days later the macrophages were lysed and serial dilutions were plated to quantify colony forming units (CFU) of viable BCG. Shown is the mean CFU from four biological replicates ± SD * *p* < 0.05 ** *p* < 0.01. Data are representative of four independent experiments. (**B**) Following IV infection with BCG (1 × 10^6^ bacteria) the total bacterial burden (expressed in CFU, mean ± SD) was determined in the spleen of wild type of *Gbp^chr3−/−^* mice 50 days following infection. Representative of two independent experiments with 4–5 mice per group, * *p* < 0.05 ** *p* < 0.01.

**Figure 2 ijms-24-02861-f002:**
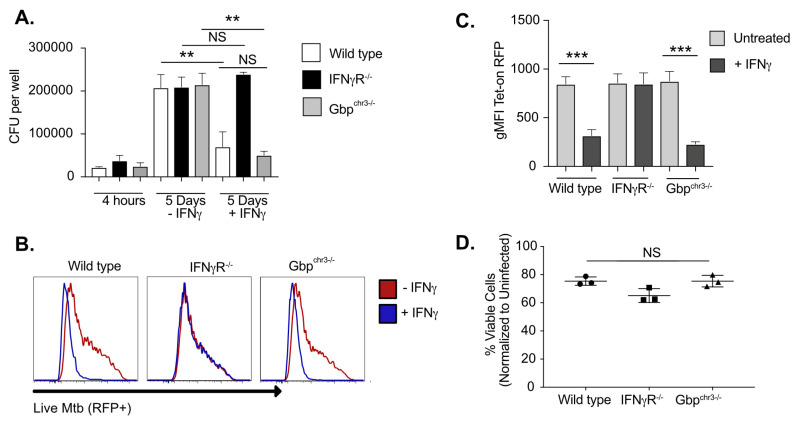
*Mycobacterium tuberculosis* is resistant to chromosome 3 GBP-mediated control in macrophages. (**A**) BMDMs from wild type, *IFNγR^/−^* or *Gbp^chr3−/−^* mice were infected with *M. tuberculosis* for 4 h then washed with fresh media in the presence or absence of IFNγ. Five days later the macrophages were lysed, serial dilutions were plated to quantify colony forming units (CFU) of viable *M. tuberculosis*. Shown is the mean CFU from four biological replicates ± SD ** *p* < 0.01; NS is not significant. Data are representative of five independent experiments. (**B**,**C**) BMDMs from wild type, *IFNγR^/−^* or *Gbp^chr3−/−^* mice were infected with the live/dead reporter *M. tuberculosis* for 4 h then washed with fresh media in the presence or absence of IFNγ. Four days later aTc was added to each well at a final concentration of 500 ng/mL. The following day infected cells were lifted and the fluorescence intensity of the inducible Tet-on TagRFP was determined by flow cytometry. Cells were gated on live and infected (mEmerald+) cells. (**B**) A representative histogram of tagRFP fluorescence is shown. (**C**) Shown is the mean fluorescence intensity of tagRFP for three biological replicates. (**D**) Cell viability was quantified from *Mtb* infected cells and normalized to uninfected cells a day five. These data are representative of three independent experiments. *** *p* < 0.001.

**Figure 3 ijms-24-02861-f003:**
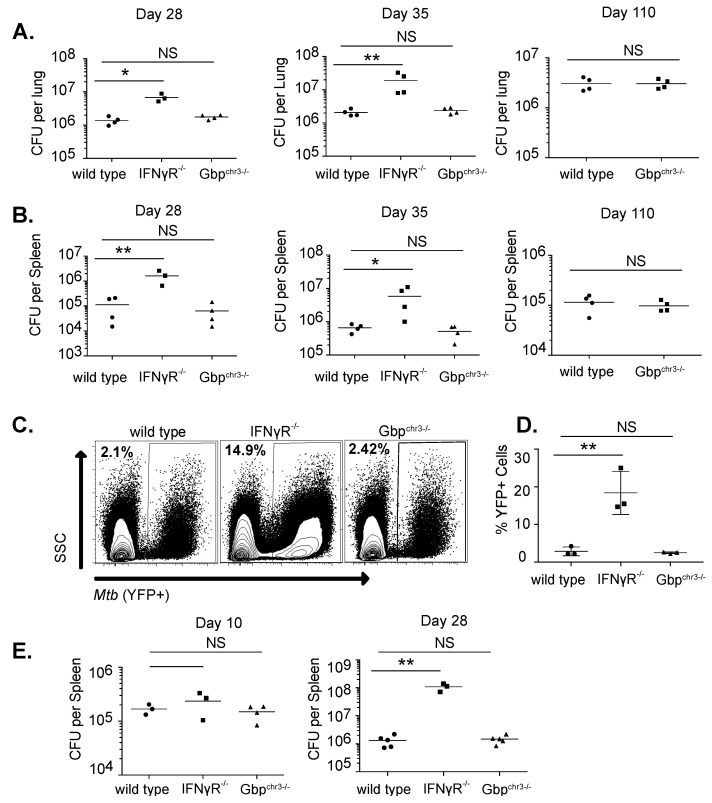
Mice lacking chromosome 3 GBPs control *M. tuberculosis* infection. Following low-dose aerosol infection with H37Rv (day 0 of ~50–150 CFU), total bacteria (expressed in CFU, mean ± SD) was determined in the lungs (**A**) or the spleen (**B**) of wild type, IFNγR^−/−^ or Gbp^chr3−/−^ mice at the indicated time points with four mice per group. 28 days following low-dose aerosol infection with sfYFP H37Rv, infected cells in the lungs of wild type, IFNγR^−/−^ or Gbp^chr3−/−^ mice were quantified by flow cytometry. One IFNγR^−/−^ mouse was euthanized prior to infection resulting in one group of three for replicate shown. (**C**) Shown is a representative flow cytometry plot of the infected cells (YFP positive) that are gated on live, single cells and (**D**) the mean percent of YFP positive cells in the lungs of infected animals. (**E**) Following intravenous infection with H37Rv (1 × 10^6^ per mouse), total bacteria were determined in the spleen of wild type, IFNγR^−/−^ or Gbp^chr3−/−^ mice at the indicated time points with 3–5 mice per group. All data are representative of 3 independent experiments. * *p* < 0.05, ** *p* < 0.01, NS not significant.

**Figure 4 ijms-24-02861-f004:**
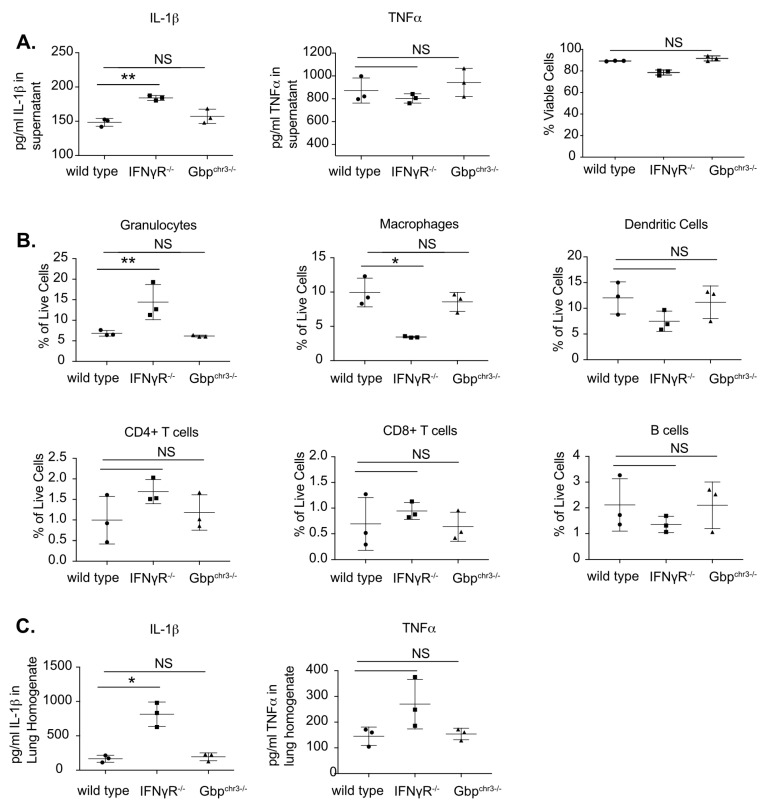
Chromosome 3 GBPs do not control the host response to *M. tuberculosis* infection. (**A**) BMDMs from wild type, *IFNγR^/−^* or *Gbp^chr3−/−^* mice were stimulated with IFNγ overnight then infected with *Mtb* for 4 h then washed with fresh media. 18 h later, (Left) supernatants were harvested and the levels of IL1β and TNFα were quantified in the supernatants by ELISA. Shown is the mean of four biological replicates normalized to a standard curve ± SD ** *p* < 0.01, NS no significance. (Right) cells viability was quantified and normalized to uninfected IFNγ-activated cells. Shown is the mean of three biological replicates ± SD. (**B**,**C**) 28 days following low-dose aerosol infection with H37Rv (day 0 of ~50–100 CFU), the lungs of wild type, *IFNγR^/−^,* and *Gbp^chr3−/−^* mice were harvested and the immune cell populations were quantified by flow cytometry and cytokines were quantified by ELISA. For (**B**), we determined the % of live cells for Granulocytes (CD45^+^ CD11b^+^ Ly6G^+^), Macrophages (CD45^+^ CD11b^+^ Ly6G^−^), Dendritic Cells (CD45^+^ CD11b^−^ Ly6G^−^ CD11c^+^), CD4^+^ T cells (CD45^+^ CD4^+^), CD8^+^ T cells (CD45^+^ CD8^+^) and B cells (CD45^+^ B220^+^). For (**C**) Cytokines quantified from lung homogenates. Shown is the mean of three mice per group ± SD * *p* < 0.05 ** *p* < 0.01, NS no significance. These data are representative of 3–4 independent experiments with similar results.

**Figure 5 ijms-24-02861-f005:**
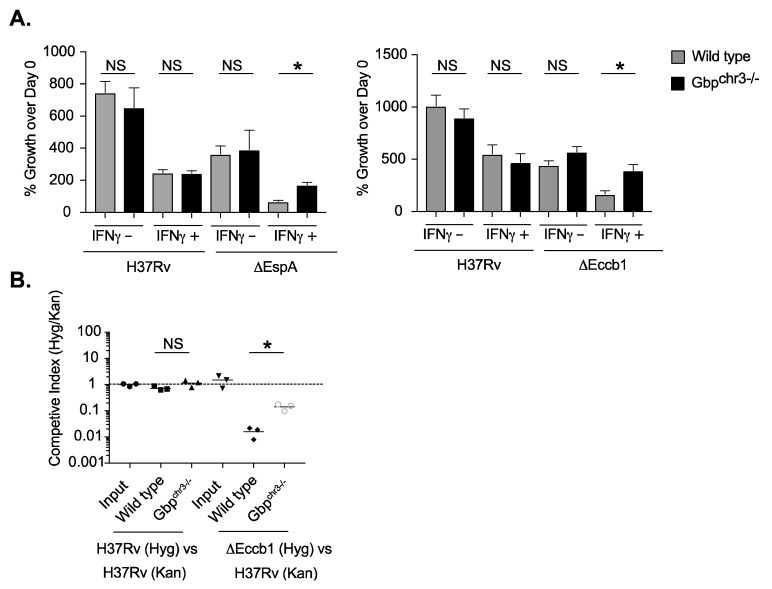
*M. tuberculosis* ESX1 mutants are susceptible to GBP-mediated control. (**A**) BMDMs from wild type or Gbp^chr3−/−^ mice were infected with *M. tuberculosis* H37Rv, ΔespA (Left) or Δeccb1 (Right) for 4 h then washed with fresh media in the presence or absence of IFNγ. Five days later the macrophages were lysed and serial dilutions were plated to quantify colony forming units (CFU) of viable *M. tuberculosis*. Shown is the mean CFU from four biological replicates ± SD * *p* < 0.05, NS no significance by two-tailed t-test. Data are representative of two independent experiments with similar results. (**B**) Following low-dose aerosol infection with a 1:1 mixed infection of either H37Rv (Hyg):H37Rv (Kan) or Δeccb1 (Hyg): H37Rv (Kan) (day 0 of ~100–200 CFU), total bacteria in the lungs that were either Kan or Hyg resistant were quantified. The competitive index was calculated (Hyg CFU/Kan CFU) and is shown as the mean for 3 independent mice for each genotype combination. The data are representative of two independent experiments with similar results.

## Data Availability

All data are included in the manuscript and further available upon request. No additional supplemental data was used in this study.

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
