# Peer review of "Mycobacterium tuberculosis Evasion of Guanylate Binding Protein-Mediated Host Defense in Mice Requires the ESX1 Secretion System"

_ijms, 2023, doi:10.3390/ijms24032861_

Round 1

Reviewer 1 Report

The protective role of IFNg in Mtb infection is well established, however, the effector pathways conferring the protection of Mtb remain unclear. In this manuscript, the authors investigated the contribution of IFN-inducible GBPs during Mtb infection, although GBPs have been shown to protect infections of various pathogens via inflammasome, autophagy, etc. The authors convincingly demonstrated that GBPs contributed to control of M.bovis BCG infection, but failed to restrict the growth of Mtb  in vitro and in vivo. Further, by employing different Mtb ESX1 mutants the authors elucidated that ESX1 could evade the protection by GBPs, which has not been reported before. In general, this study provided novel insights on the roles of GBPs in mycobacterial infection and Mtb evasion of the defense by GBPs. Some minor comments are detailed below.

1.    For Fig. 2, the authors should investigate the involvement of GBPs in Mtb-induced pyroptotic cell death to exclude the possibility of different cell numbers, though the author measured IL1B by ELISA in Fig.4.

2.    For Fig. 3A&3B, there are only 3 dots for IFNgR KO group. Please state the reason of excluding one mice in the manuscript.

3.    The stars indicating statistical significance are missing in Fig. 4.

4.    For Fig. 5, Please provide statistical analysis of groups between Mtb WT and mutant strains.

Author Response

Reviewer 1

The protective role of IFNg in Mtb infection is well established, however, the effector pathways conferring the protection of Mtb remain unclear. In this manuscript, the authors investigated the contribution of IFN-inducible GBPs during Mtb infection, although GBPs have been shown to protect infections of various pathogens via inflammasome, autophagy, etc. The authors convincingly demonstrated that GBPs contributed to control of M.bovis BCG infection, but failed to restrict the growth of Mtb  in vitro and in vivo. Further, by employing different Mtb ESX1 mutants the authors elucidated that ESX1 could evade the protection by GBPs, which has not been reported before. In general, this study provided novel insights on the roles of GBPs in mycobacterial infection and Mtb evasion of the defense by GBPs. Some minor comments are detailed below.

We thank the reviewer for the positive comments and summary of our manuscript.

  1. For Fig. 2, the authors should investigate the involvement of GBPs in Mtb-induced pyroptotic cell death to exclude the possibility of different cell numbers, though the author measured IL1B by ELISA in Fig.4.

We thank the reviewer for pointing out this possibility. We have added the requested cell death data to the manuscript. We completed a cell titer glo viability assay at 24 hours and five days following infection. We noted no significant differences in cell viability between wild type and GBPchrm3-/- BMDMs suggesting these GBPs do not play a major role in modulating cell death response to M. tuberculosis infection. We have included these data in revised versions of Figure 2 and Figure 4.

  1. For Fig. 3A&3B, there are only 3 dots for IFNgR KO group. Please state the reason of excluding one mice in the manuscript.

Thank you for pointing out this discrepancy. Prior to the infection of this cohort of animals one IFNgR KO mouse needed to be euthanized for health reasons. Thus, there was one less mouse for this genotype in this time course experiment. We have stated this explicitly in the figure legend for clarity.

  1. The stars indicating statistical significance are missing in Fig. 4.

We apologize for this error and have double checked the figures statistical representations in the revised figure.

  1. For Fig. 5, Please provide statistical analysis of groups between Mtb WT and mutant strains.

We have added these comparisons to the figure.

Reviewer 2 Report

This is a great paper and important for the field given the excitement about GBPs in other pathogen infections. It's important that these BCG versus Mtb differences are made clear to the field. Strengths of the paper include using multiple measurement modalities and experimental tools to buttress the scientific conclusions of the manuscript. One experiment to suggest to the authors for the future is to ask if GBPs are even recruited to the Mtb-containing phagosome. In the discussion, is there anything known about the cell biology of GBPs which might help explain the differences observed? I note they talk about ESX-1 mediated evasion however is there anything more specific the authors could say here? It seems like GBPs are ubiquitinated following Mtb infection in mouse macrophages...could an alternative mechanism be differences in ubiquitination of GBPs between Mtb and ESX-1 deficient mycobacteria?

Author Response

Reviewer 2

This is a great paper and important for the field given the excitement about GBPs in other pathogen infections. It's important that these BCG versus Mtb differences are made clear to the field. Strengths of the paper include using multiple measurement modalities and experimental tools to buttress the scientific conclusions of the manuscript.

Thank you for the positive comments about the manuscript and the rigor of our results.

One experiment to suggest to the authors for the future is to ask if GBPs are even recruited to the Mtb-containing phagosome. In the discussion, is there anything known about the cell biology of GBPs which might help explain the differences observed? I note they talk about ESX-1 mediated evasion however is there anything more specific the authors could say here? It seems like GBPs are ubiquitinated following Mtb infection in mouse macrophages...could an alternative mechanism be differences in ubiquitination of GBPs between Mtb and ESX-1 deficient mycobacteria?

The reviewer brings up an excellent question regarding the mechanisms underlying differences in GBP-mediated control in the presence and absence of ESX-1. While we have no direct experimental evidence towards this, we have now expanded our discussion of potential mechanisms. The next key experiments needed are to determine if the expression or stability of GBPs are altered during Mtb infections with and without ESX-1. In addition, as the reviewer points out, understanding if GBPs are differentially recruited to the Mtb containing vacuole in the presence and absence of ESX-1. These experiments will help to better understand how Mtb avoids GBP-mediated control. If expression is altered it suggests Mtb actively modulates GBP expression, if GBP recruitment and modification to the Mtb containing vacuole is changed it suggests ESX-1 directly contributes to remodeling the cell-autonomous immune response during infection. We have now added these models and speculation on potential mechanisms to the discussion of the revised manuscript.